# BOOTSTRAPPED HINDSIGHT EXPERIENCE REPLAY WITH COUNTERINTUITIVE PRIORITIZATION

## ABSTRACT

Goal-conditioned environments are known as sparse rewards tasks, in which the agent gains a positive reward only when it achieves the goal. Such an setting results in much difficulty for the agent to explore successful trajectories. Hindsight experience replay (HER) replaces the goal in failed experiences with any practically achieved one, so that the agent has a much higher chance to see successful trajectories even if they are fake. Comprehensive results have demonstrated the effectiveness of HER in the literature. However, the importance of the fake trajectories differs in terms of exploration and exploitation, and it is usually inefficient to learn with a fixed proportion of fake and original data as HER did. In this paper, inspired by Bootstrapped DQN, we use multiple heads in DDPG and take advantage of the diversity and uncertainty among multiple heads to improve the data efficiency with relabeled goals. The method is referred to as Bootstrapped HER (BHER). Specifically, in addition to the benefit from the Bootstrapped version, we explicitly leverage the uncertainty measured by the variance of estimated Q-values from multiple heads. A common knowledge is that higher uncertainty will promote exploration and hence maximizing the uncertainty via a bonus term will induce better performance in Q-learning. However, in this paper, we reveal a counterintuitive fact that for hindsight experiences, exploiting lower uncertainty data samples will significantly improve the performance. The explanation behind this fact is that hindsight relabeling itself largely promotes exploration, and then exploiting lower uncertainty data (whose goals are generated by hindsight relabeling) provides a good trade-off between exploration and exploitation, resulting in further improved data efficiency. Comprehensive experiments demonstrate that our method can achieve state-of-the-art results in many goal-conditioned tasks.

## 1 INTRODUCTION

Deep reinforcement learning (DRL) has gained significant achievements in solving games (Zha et al., 2019; Crespo & Wichert, 2020; Oh et al., 2021) and robotics problems (Rahimi et al., 2018; Nguyen & La, 2019; Zhu et al., 2021). Despite these successes, sparse rewards problems are still challenging. In sparse reward tasks, the agent gets a non-negative reward only when it achieves some goal. As a consequence, it is very difficult for an agent to see successful trajectories. Indeed, many state-of-the-art RL algorithms, such as TRPO (Schulman et al., 2015), PPO (Schulman et al., 2017), DQN (Mnih et al., 2015), and DDPG (Lillicrap et al., 2019), etc., often fail to perform well in sparse rewards environments.

To solve sparse rewards problems, there have been rich literature devoted to reward engineering. Potential Based Reward Shaping (PBRS) (Ng et al., 1999) uses difference between two potential functions to design a shaping function, where the potential function comes from the expertise knowledge and evaluates the value of state. Potential Based state-action Advice (PBA) (Wiewiora et al., 2003) extend PBRS to use potential function to evaluates the value of state-action pair. Dynamic Potential Based (DPB) (Devlin & Kudenko, 2012) approach takes time as one of the inputs of the potential function. However, reward engineering usually requires domain-specific knowledge and it often suffers from sub-optimal performance. A brilliant idea for solving goal-conditioned tasks is using hindsight experiences. HER (Andrychowicz et al., 2017), which replaces the desired goal in failed experiences with some achieved state in the failed trajectory to pretend that the agent obtains a positive reward, has been demonstrated its effectiveness in a wide range of goal-conditioned appli-

cations. However, HER has its own limitations that when relabeling the goal in one failed trajectory, there exists many choices of the pseudo goals that any achieved state can be selected. Generally, uniformly sampling one of them, as HER did, is not the most effective way from the perspective of importance sampling.

In this paper, we focus on improving the data efficiency in HER via the following two directions: 1) inspired by Bootstrapped DQN (Osband et al., 2016), we propose a bootstrapped version of HER that employs multiple Q-value heads to increase the diversity in choosing the pseudo goals for deep exploration. In this paper, we use DDPG as the basic RL algorithm, while the proposed techniques can be integrated by any off-the-shelf RL algorithms; 2) in addition to the benefit on exploration by using the bootstrapped Q-learning, we further explicitly leverage the uncertainty among the multiple heads by sampling goals conditioning on the variance of the estimated Q-values. Many previous approaches have shown that data with higher value uncertainty can promote exploration, and maximizing the uncertainty by incorporating a bonus term in the reward has been widely considered, such as bandit algorithms (Lattimore & Szepesvári, 2020). To our surprise, this is totally not the case in HER. On the contrast, sampling goals with lower value uncertainty first is demonstrated to be significantly better than uniform sampling, and sampling goals with high uncertainty in prior deteriorates the performance of HER a lot. We thus call this empirical fact as counterintuitive prioritization. Indeed, this can be explained by the trade-off between exploration and exploitation when importing HER, bootstrapped Q-learning and importance sampling. We refer our algorithm to as the Bootstrapped HER with counterintuitive prioritization (BHER). We experiment with a number of control tasks with continuous action space. The experimental results verify the superiority of BHER over many state-of-the-art baseline methods.

## 2 RELATED WORK

Sparse rewards in goal-conditioned environment is one of the major challenges of RL at present. The earlier work is UVF (Schaul et al., 2015) algorithm, which uses state and goal together as the conditions for agent to take action. Bootstrapped DQN (Osband et al., 2016) uses the bootstrap principle to expand network structure. It uses multiple randomized value functions to estimate the Q-value of the state-action pair. This is equivalent to agent having multiple policies to interact with the environment, which effectively enhances agent's exploration. But these methods don't work well in the environment of sparse rewards. Hindsight Experience Replay (HER) (Andrychowicz et al., 2017) algorithm recalculates reward by randomly replacing the desired goal in the failed transition with the achieved goal. In this way, pseudo successful data can be obtained, which can improve data efficiency. HER algorithm better solves the problem of sparse rewards.

Since then, there have been many studies related to HER algorithm. Dynamic Hindsight Experience Replay (DHER) (Fang et al., 2019a) utilizes the HER algorithm to process dynamic goals in the environment. Competitive Experience Replay (CER) (Liu et al., 2019) uses the competition between two agents to obtain better exploration. Goal-Conditioned Supervised Learning (GCSL) (Ghosh et al., 2020) is a kind of imitation learning. It uses successful trajectories generated by agent as expert demonstrates, and use supervised learning to optimize a lower bound on the goal-oriented RL objective. PlanGAN (Charlesworth & Montana, 2020) is a model-based RL algorithm that uses GANs (Creswell et al., 2018) to learn the model of the environment and make planning. But this method requires a huge amount of computation to select a single action.

Recently, researchers have realized that not all data have the same value to learn (Schulman et al., 2017). More of the work is to further improve data efficiency by adding priority replay on HER algorithm. Energy-Based HER (Zhao & Tresp, 2018) calculates the kinetic energy, potential energy and rotational energy as the total energy of the achieved goal according to the energy principle in physics. Then it gives higher priority to transitions with larger energy. Maximum Entropy-based Prioritization (MEP) (Zhao et al., 2020) calculates the entropy of a trajectory which is used as priority. Curriculum-guided HER (CHER) (Fang et al., 2019b) calculates the proximity and diversity between the achieved goal and the desired goal by designing two functions, which are added together with a weight to balance exploration and exploitation. These methods can be regarded as a form of curriculum learning (Elman, 1993; Bengio et al., 2009; Zaremba & Sutskever, 2014; Graves et al., 2017). But these algorithms either make some assumptions about the environment or design some specific functions to calculate the priority. Unlike the above works, our BHER algorithm does not

have any requirements on the environment when calculating the priority, and it is easy to generalize to other goal-conditioned environments.

## 3 PRELIMINARIES

In this section, we introduce the preliminaries, such as the challenge of goal-conditioned environment, HER (Andrychowicz et al., 2017) and Bootstrapped DQN (Osband et al., 2016) which motivate us to propose our BHER algorithm.

### 3.1 GOAL-CONDITIONED ENVIRONMENT

In RL, the problem considered is usually composed of an agent and an environment. At time step $t$, the agent takes action $a_t$ conditioned on the state $s_t$ of the environment. After the environment receives the action $a_t$, it transfers to the state $s_{t+1}$ and feeds back the reward $r_t$. The agent learns a policy $\pi(a|s)$ to maximize expected discounted return:

$$G = \mathbb{E}_\tau \left[ \sum_{t=0}^{T-1} \gamma^t r_t \right], \tag{1}$$

where $\tau$ is a trajectory $(s_0, a_0, r_0, ..., s_{T-1}, a_{T-1}, r_{T-1})$ and $\gamma$ is the discount factor.

In goal-conditioned environment, the agent takes an action $a$ by taking an additional input, i.e., a goal $g$, and the policy is hence a function of $(s, g)$ that $a \sim \pi(\cdot|s, g)$. Generally, the rewards in goal-conditioned environments are sparse, and only when the agent reaches the goal, it gets a non-negative feedback. Therefore, it is difficult for the agent to explore successful trajectories when its policy starts from scratch.

### 3.2 HER AND BOOTSTRAPPED DQN

HER replaces the *desired goal* $g$ in the failed transition with some sampled *achieved state* $g'$ in the current trajectory, and then recalculates the reward based on the pseudo goal $g'$. In this way, the agent can see successful trajectories much more frequently, even if the goal is fake. It has been demonstrated in rich literature that HER is very effective for solving goal-conditioned tasks and it can be adopted by any off-policy algorithms, such as DQN (Mnih et al., 2015), DDPG (Lillicrap et al., 2019), TD3 (Fujimoto et al., 2018), etc.

Bootstrapped DQN designs multiple value function heads which are initialized randomly in the network structure of the DQN algorithm, and all heads share the underlying network. Each time agent interacts with the environment, one of the value function heads is randomly selected as policy to generate a trajectory. Each value function head is updated according to its own target network for temporally extended exploration, and each value function head only uses data generated by itself for updates. Bootstrapped DQN subtly applies the Thompson sampling (Russo et al., 2017) heuristic to RL which allows for deep exploration.

## 4 METHODOLOGY

In this section, we introduce the bootstrapped version of HER with multiple Q-value heads and the counterintuitive prioritization.

### 4.1 BOOTSTRAPPED HER

We use DDPG as the basic RL algorithm. In goal-conditioned tasks, the policy and Q-value take both the state and goal as input. A bootstrapped version of DDPG is dipicted in Fig. 1, where both the actor network and critic network are implemented with multiple heads. All critic heads share an underlying embedding network, similar to what was used in Bootstrapped DQN (Osband et al., 2016), and actor heads share an underlying network for feature extraction as well. Each actor head is updated according to its corresponding critic head, for which its pseudo goals are sampled independently and it is updated by according to its own recomputed rewards.

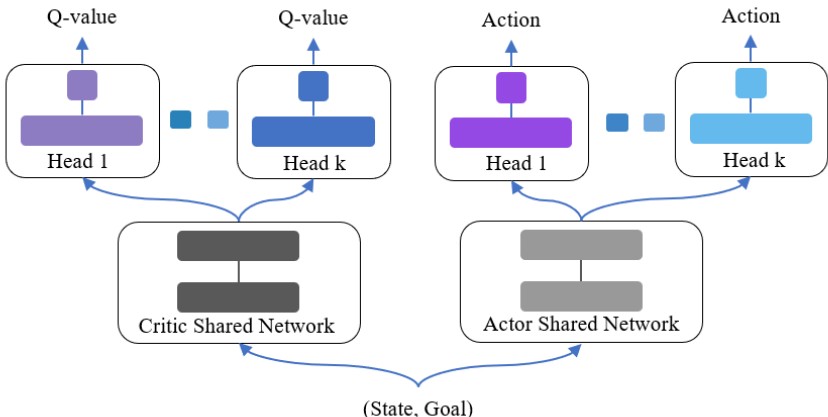

Figure 1: The network structure of Bootstrapped HER.

As has been demonstrated in Bootstrapped DQN, random initialization of the parameters of each head is sufficient to generate diversity for deep exploration. This is also verified in our experiments for BHER. In this way, we can benefit a lot in terms of exploration with the cost of increasing only a small amount of the network parameters from multiple heads, since they share a large proportion of parameters in the embedding network.

In Appendix D, we provide an illustration of the distribution of goals the agent can arrive in a toy environment by using multiple heads. Compared to the standard HER, we can verify that the bootstrapped version of HER is much more efficient in terms of exploration.

## 4.2 COUNTERINTUITIVE PRIORITIZATION

As we have shown, employing multiple Q-value heads for HER can effectively promote exploration, since each head is updated by sampling its own trajectories and pseudo goals. Moreover, with multiple critic heads, it is natural to compute the variance among the estimated Q-values. In this section, we propose to explicitly leverage this information. Specifically, given a transition, we inference all the Q-values and calculate the variance, which is stored in the replay buffer. Then, at the back-propagation stage, we sample the mini-batch conditioning on the stored variance. A common experience is that sampling the data with higher variance will further encourage exploration on these transitions that the policy rarely see. Previous results (Pathak et al., 2017; Zhelo et al., 2018; Li et al., 2020) have demonstrated this principle can result in much better performance in RL. However, as we will explain below and verify in the experiments, this principle is never true for BHER.

To verify this, we use the Reacher environment (for more details, please refer to Appendix A) as a toy example to visualize the variance of the Q-values stored in data. Fig. 2 shows that during the training process at different training epochs, the variances of the hindsight transitions are consistently smaller than the variances of the original transitions. This is explainable, because in hindsight transitions, the agent always receives positive (pseudo) rewards, while for the original transitions without goal relabeling, the agent fails to reach the goal at most of the time at the early training epochs, unless the policy is good enough. Indeed, in Fig. 2(c), when we have trained over more training epochs, the variance in the original transitions decrease. According to RL, the policy will be updated more aggressively over the transitions whereas the values are higher. Therefore, the hindsight experiences deserve a lower variance of the Q-values in BHER.

Now, it becomes straightforward that if we give high priority in sampling transitions with larger Q-value variances, we are still sampling original transitions without goal relabeling. That is, we are not performing HER any more. Therefore, we should assign higher priority to hindsight transitions with smaller Q-value variances in the training process. That is, we enhance the exploitation on the

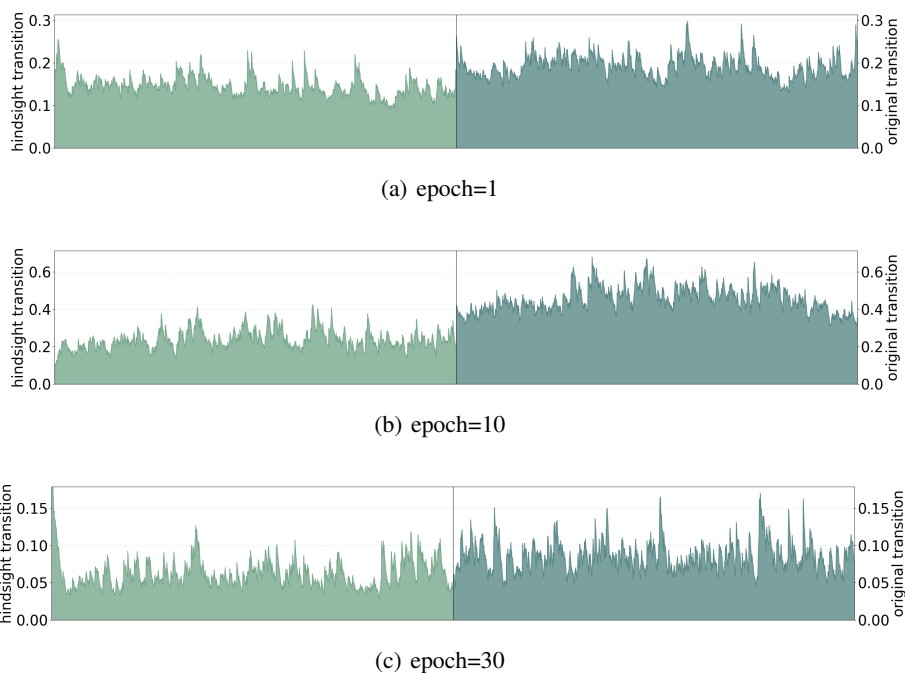

(a) epoch=1

(b) epoch=10

(c) epoch=30

Figure 2: Variance of the Q-values estimated from multiple heads in BHER on Reacher. The left side denotes the variances of the hindsight transitions (with modified goals) in the replay buffer, while the right side plots the variances of the original transitions. In this illustration experiment, the *ratio* of sampled original data is set to $50\%$, and another half of data is composed by the hindsight samples (in the original work of HER, this ratio is set to $20\%$ by default). This is to eliminate the impact of data imbalance on evaluating the variances in the data. In our formal experiments, this ratio keeps the value as the same in HER.

hindsight experiences by sampling these data more aggressively. So, we refer this sampling principle to as *Counterintuitive Prioritization*.

Indeed, BHER with Counterintuitive Prioritization provides a trade-off between exploration and exploitation that the bootstrapped Q-learning encourages deep exploration and more diversity, while counterintuitive prioritization promotes exploitation in the data samples which are selected by the bootstrapped progress and are worthy exploration. As we will demonstrate in our experiments on a wide range of environemnts, the above methods can improve the standard HER method by a large margin in performance.

### 4.3 BOOTSTRAPPED HER WITH COUNTERINTUITIVE PRIORITIZATION

In this section, we provide a detailed description of the proposed algorithm. In the following context, the abbreviation BHER denotes the bootstrapped HER with counterintuitive prioritization by default.

---

**Algorithm 1** Calculate Priority

---

1: **Requirement:** original buffer $\mathcal{B}_o$ and HER buffer $\mathcal{B}_h$
2: **while** training is not ending **do**
3:     Sample minibatch transitions $B_o$ from $\mathcal{B}_o$
4:     Replace desired goals with achieved goals in transitions $B_o$ with HER
5:     Calculate variance of Q-value for each transition in $B_o$
6:     Store transitions $B_o$ in $\mathcal{B}_h$ with variances
7:     Update priority for each transition in $\mathcal{B}_h$ (see Equation(2))
8: **end while**

---

Similar to Bootstrapped DQN (Osband et al., 2016), in our implementation of BHER, all heads share most of the parameters, and each head is updated according to its own target. However, an important difference is that each head in Bootstrapped DQN has to maintain its own replay buffer to distinguish the data (diversity) generated by different heads (policies), while in BHER this is not necessary. Note that in BHER, most transitions are hindsight experiences, which are not exactly generated by the training policies, so there is no need to distinguish which head generated the transitions, and all the heads can share the same replay buffer.

To implement counterintuitive prioritization, we create two replay buffers. One replay buffer has larger size and it stores the original interactions between the agent and the environment, and we call it the *original buffer* $\mathcal{B}_o$. Another one is a smaller buffer, which stores a proportion of sampled hindsight and original data, and we call this buffer the *HER buffer* $\mathcal{B}_h$. During training, we first sample in the original buffer according to the original HER algorithm, replace the goals according to a certain proportion, and then store them in the HER buffer. We then calculate the priority of each transition $t$ in the HER buffer according to the formula:

$$p_t = \left( \frac{\sigma_{max}^2 - \sigma_t^2 + \epsilon}{\sum_{\sigma \in \mathcal{B}_h} (\sigma_{max}^2 - \sigma^2 + \epsilon)} \right)^T \qquad (2)$$

where $\sigma_t^2$ is the variance of $Q$-values of transition $t$, and $\sigma_{max}^2$ is the largest variance in the current HER buffer; $\epsilon$ is a very small number for numerical stability, and we set it to $1e^{-6}$ in our experiments; $T$ is a temperature coefficient used to adjust the sharpness of the priority distribution in the HER buffer. Then, the transitions in the HER buffer are sampled according to the calculated priority. Algorithm 1 briefly illustrates the implementation of priority calculation process in our algorithm.

For data generation, each time we use all the actor heads to generate multiple trajectories and store them in the original buffer to ensure the diversity of the original data. When sampling data in the original buffer, we use the 'future' method in the HER algorithm, and set replay times to $4$. That is, $20\%$ of the data in the HER buffer is original data. Algorithm 2 briefly illustrates the implementation of our algorithm. In our experiments, we set the number of heads to $8$, which is sufficient to allow the agent explore well. The detailed settings of other hyperparameters are reported in Section 5.3.

---

**Algorithm 2** Bootstrapped Hindsight Experience replay with Counterintuitive Prioritization

---

1: **Initiate:** an off-policy RL Algorithm $\mathbb{A}$, original buffer $\mathcal{B}_o$ and HER buffer $\mathcal{B}_h$
2: **for** training epoch $= 0, 1, 2, \cdots, M - 1$ **do**
3:    **for** cycle $i = 0, 1, \cdots, N - 1$ **do**
4:       **for** head $k = 0, 1, \cdots, K - 1$ **do**
5:          Sample an initial goal $g$ as desired goal and initial state $s_o$
6:          **for** timestep $t = 0, 1, \cdots, T - 1$ **do**
7:             Task an action $a_t$ from the behavioral policy of $\mathbb{A}$, i.e., $a_t \sim \pi_k(\cdot|s_t, g)$
8:             Observe a new state $s_{t+1}$ and achieved goal $g_t'$
9:             Calculate reward $r_t := r(s_t, a_t, g)$
10:            Store the transition $(s_t, a_t, r_t, g, g_t')$ in $\mathcal{B}_o$
11:          **end for**
12:       **end for**
13:    Sample transitions $B_o$ from $\mathcal{B}_o$ and store them in $\mathcal{B}_h$ by Algorithm 1, i.e.,
14:       **for** head $k = 0, 1, \cdots, K - 1$ **do**
15:          Sample a minibatch transitions $B_h$ according to priority in $\mathcal{B}_h$
16:          Optimize $k$-th head of $\mathbb{A}$ using minibatch transitions $B_h$
17:       **end for**
18:    **end for**
19:    Evaluate the average performance of all heads through 100 rollouts
20: **end for**

---

## 5 EXPERIMENT

We use a number of goal-reaching and robotics control environments to evaluate our BHER algorithm and compare with state-of-the-art baselines. We first introduce the environment and our

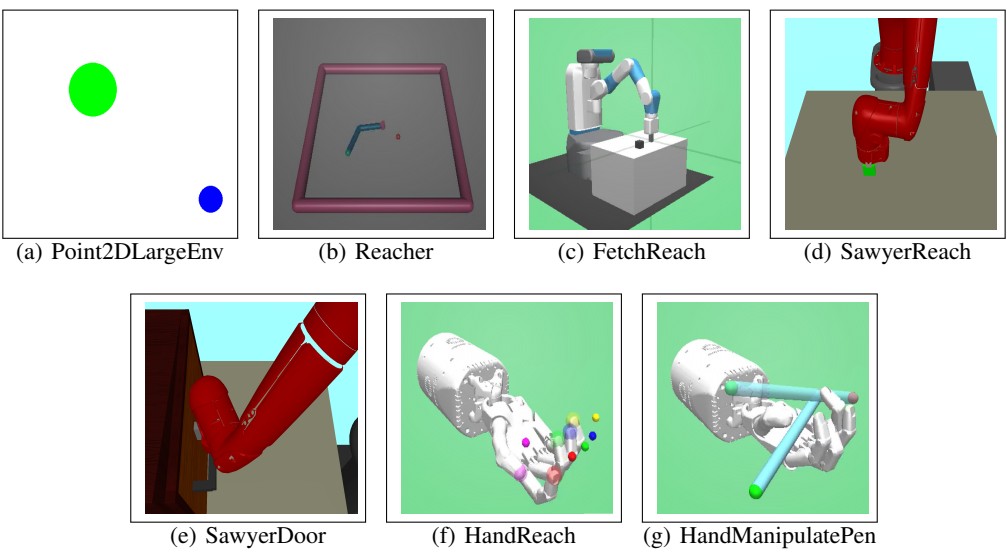

|                       |             |               |                |
|:---------------------:|:-----------:|:-------------:|:--------------:|
| (a) Point2DLargeEnv   | (b) Reacher | (c) FetchReach | (d) SawyerReach |

| | | |
|:---:|:---:|:---:|
| (e) SawyerDoor | (f) HandReach | (g) HandManipulatePen |

Figure 3: Environment.

comparison results. In the end, we adjust and compare our own algorithms and show the results of ablation experiments.

## 5.1 ENVIRONMENTS

Fig. 3 shows the goal-conditioned environments we have evaluated in our experiment. Goal-reaching environments include Point2DLargeEnv, Reacher, FetchReach and SawyerReach, and robotics control tasks include SawyerDoor, HandReach and HandManipulatePen. The state space and action space of all environments are continuous, and the reward is sparse and non-negative. Only when agent reaches desired goal, it gets a reward of $0$, otherwise it receives a reward of $-1$. The maximum number of steps in each environment is set to $50$. For a more detailed description of the environments, please refer to Appendix A.

## 5.2 BASELINES

The implementation of all algorithms is based on the DDPG algorithm. We compare our algorithm with the following state-of-the-art HER based methods:

- HER (Andrychowicz et al., 2017), which does not use prioritization and randomly samples transitions.
- HEREBP (Zhao & Tresp, 2018), which calculates the energy of transitions and use it as the sample priority.
- CHER (Fang et al., 2019b), which calculates the proximity and diversity of transitions as the sample priority.

We compare our algorithms with these baselines which are implemented by their open-sourced codes in all the environments described above.

## 5.3 COMPARISON RESULTS ON ALL ENVIRONMENTS

The common hyperparameters of all algorithms are set as the same. When we experiment in the goal-reaching environments, we set the epoch to $30$ and the batch size to $64$. In the robotics control tasks, the epoch is set to $100$ and the batch size is $256$. We do not use MPI to parallelize the program on multiple workers. In order to ensure that all algorithms have the same amount of data during training, our algorithm uses $8$ heads to generate multiple trajectories each time, while other

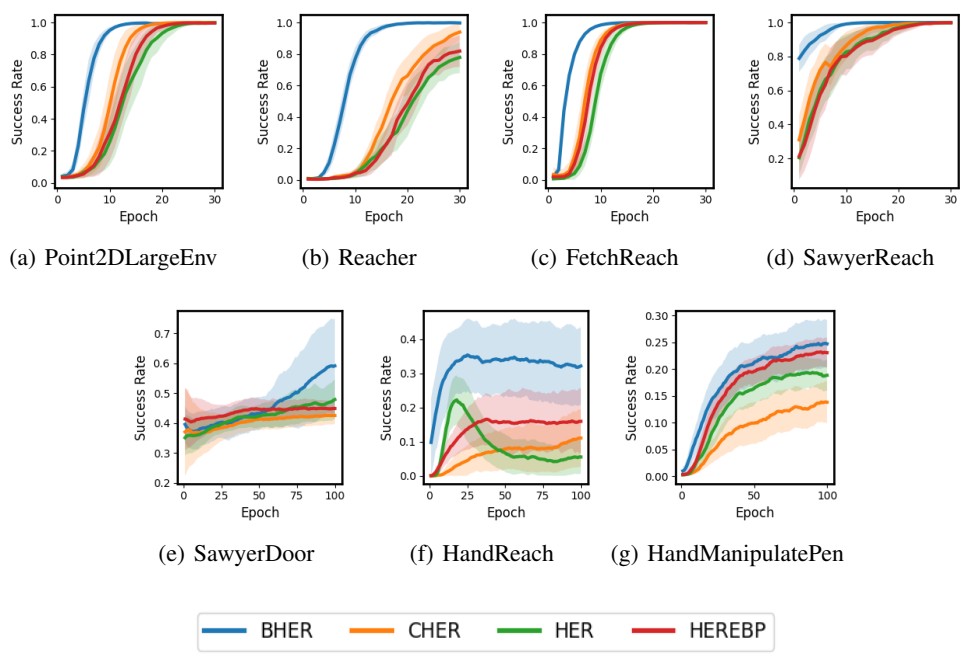

(a) Point2DLargeEnv (b) Reacher (c) FetchReach (d) SawyerReach

(e) SawyerDoor (f) HandReach (g) HandManipulatePen

Figure 4: Performance for allenvironments: success rate (line) with standard deviation range (shaded area) on all environments acorss 10 random seeds.

algorithms use DummyVecEnv to run 8 environments in parallel. After each epoch, we perform 100 rollouts for evaluation, and calculate the average success rate of the 100 rollouts. In our algorithm, each head generates a trajectory during each rollout progress when evaluation.

Fig. 4 shows the comparison results of all algorithms. For each algorithm, we use 10 random seeds to repeat the training and testing and plot the mean curve and the standard derivation confidence. It can be observed that our algorithm outperforms all other baselines. In some environments, the performance of HEREBP and CHER is not as good as HER, which may be because the method of calculating priority in these two algorithms is not suitable for the current environment. For example, in SawyerDoor environment, the goal has no potential energy or rotational energy, and HEREBP algorithm can only use kinetic energy as the priority of transition.

For BHER, we fine tune different temperature coefficients in calculating the priority for each environment in our algorithm, as shown in Appendix B. This hyperparameter can slightly adjust the performance of our algorithm. We choose the best temperature coefficient for each environment for BHER algorithm.

## 5.4 ABLATION EXPERIMENTS RESULTS

We conducted ablation experiments to confirm the effectiveness of counterintuitive prioritization. Section 4.2 illustrates that counterintuitive priority replay method (giving higher priority to smaller variances of Q-value) can improve the exploitation of algorithm. Here we compare it with a variant of our BHER algorithm that only uses Bootstrapped DDPG and do not use counterintuitive prioritization, referred to as BHER w/o prioritization. In addition, we also compared the way of prioritizing transitions based on higher variance of Q-value which we call BHER w/ higher variance first.

The result in Fig. 5 verifies the effectiveness of counterintuitive prioritization. It improves the exploitation of algorithm by prioritizing transitions with small variance of the Q-value, and further improves data efficiency.

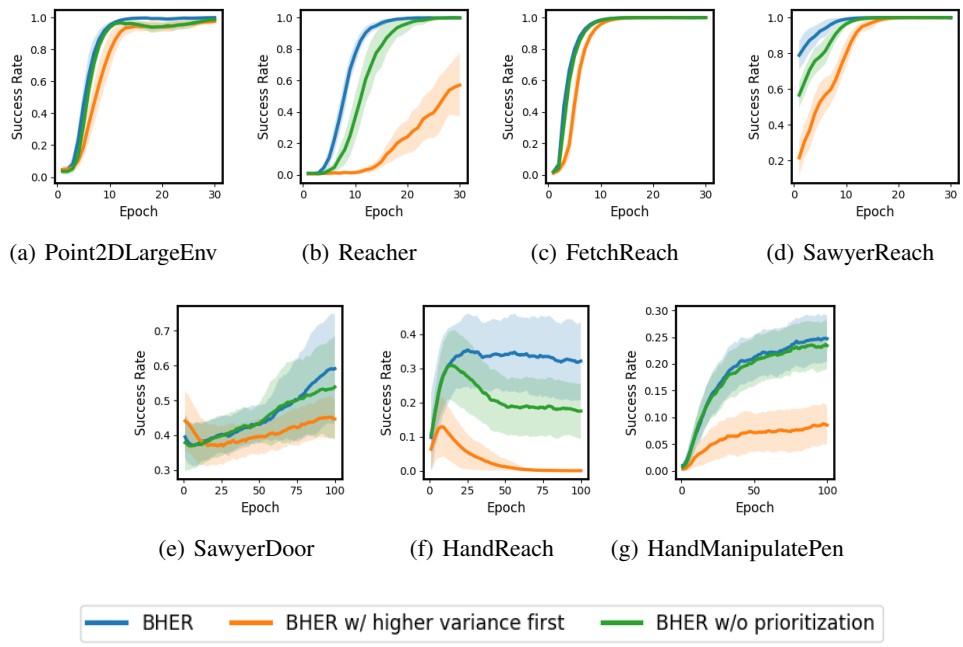

Figure 5: Ablation experiments: success rate (line) with standard deviation range (shaded area) on all environments acorss 10 random seeds.

We also conducted an experimental comparison on the adjustment of the priority temperature coefficient. Appendix B shows the results. It can be seen that different environments have different optimal priority temperature coefficients.

In addition, Section 4.2 illustrates that the variances of the hindsight transitions are smaller than the variances of the original transitions, and we give higher priority in transitions with smaller variance. In this way, hindsight transitions have a higher probability of being sampled by the algorithm. Intuitively, we will think about the impact of not using original data on the performance of our algorithm. To see this, we conduct ablation experiments to evaluate the influence of the usage of the original data. We show our ablation experimental results in Appendix C. It illustrates that in some simple environments, the usage of original data has little effect on our algorithm. But in more complex environments, original data is necessary. The ratio of the original data is not the focus of our algorithm, we just set it to $20\%$ according to the original HER algorithm.

## 6 CONCLUSION

The main contributions of the BHER algorithm we proposed are as follows: 1) The way we use the bootstrapped principle to design the network has greatly improved HER algorithm's exploration in environment of sparse rewards. And when we update the network, we do not distinguish the source of the data, which greatly improves the sample efficiency. 2) We designed a counterintuitive prioritized replay method to improve the exploitation of algorithms, which provides a new criterion for priority replay in RL. And our priority calculation method will not vary depending on the environment, which makes our algorithm have better generalization. 3) We combined the bootstrapped principle and counterintuitive prioritization, which makes a excellent trade-off between exploration and exploitation in RL. 4) Whether in a simple goal-reaching environment or a difficult robotics controll environment, our algorithm has achieved state-of-the-art preformance. 5) Our algorithm does not have any assumptions or requirements on the environment, and can be easily extended to any goal-conditioned environment.

## REPRODUCIBILITY

We provide a Reproducibility Statement below. For the algorithms, we have attached the main codes of BHER in the submitted .zip supplementary material.

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

APPENDIX

## A  ENVIRONMENTS

Fig. 3(a) 3(b) 3(c) 3(e) shows the goal-reaching environment we have used, including four environments: Point2DLargeEnv, Reacher, FetchReach and SawyerReach. Point2DLargeEnv is a point-based environment, in which the blue point aims to reach the green circle. The blue point can reach any position in the environment. The state and goal of this environment are both 2-dimensional vectors, which respectively represent the position of the blue point and the green circle, and the action is also a 2-dimensional vector, which represents the moving distance of the green circle each time. In Reacher, a 2-DoF (degrees of freedom) robotic arm whose task is to reach a particular target in the field. Its state is an 11-dimensional vector, which contains the angle of the two joints and the position of the fingertip relative to the target, and goal is a 2-dimensional vector that represents the position of the target, and action is also a 2-dimensional vector, representing the movement of two joints. FetchReach and SawyerReach are two very similar environments. Both have a 7-DoF robotic arm pushing a box until it reaches the target position. In FetchReach, robotics arm has a two-fingered parallel gripper. The action is a 3-dimensional vector representing the desired gripper' movement. The goal is a 3-dimensional vector describing the position of the target and the achieved goal is the position of the gripper. In SawyerReach, the arm's end-effector (EE) is constrained to a 2-dimensional rectangle parallel to a table and constrained to only move in the 2-dimensional plane. The action controls EE position through the use of a mocap. The state is the 2-dimensional position of the EE and the goal is an 2-dimensional position of the EE.

Fig. 3(e) 3(f) 3(g) shows the robotics controll environment we have used, including three environments: SawyerDoor, HandReach and HandManipulatePenRotate. In SawyerDoor, these is a robotics arm which aims to open a small cabinet door, initially shut closed, sitting on a table to a specified angle. The state is a 4-dimensional vector consisting of the Cartesian coordinates of the arm's end-effector and the door's angle. The action is a 3-dimensional vector controlling the position of the end-effector. The goal is the desired angle at which the door is opened. Hand environments has a 24-DoF anthropomorphic robotic hand, whose 20 joints can be can be controlled independently whereas the remaining ones are coupled joints (Plappert et al., 2018). So the action is a 20-dimensional vector containing the absolute position control for all non-coupled joints of the hand. In HandReach, the goal is 15-dimensional vector and contains the target Cartesian position of each fingertip of the hand and is considered achieved if the mean distance between fingertips and their desired position is less than 1 cm. In HandManipulatePenRotate, a pen is placed on the palm of the hand. The task is to then manipulate the pen such that a target pose is achieved. The goal is a 7-dimensional vector including the target position in Cartesian coordinates and target rotation in quaternions. We use the rotate mode which includes the target rotation $x$ and $y$ axes of the pen and without target rotation around the $z$ axis or target position.

## B  OPTIMAL PRIORITY TEMPERATURE COEFFICIENTS

We evaluate the impact of different priority temperature coefficients on our BHER algorithm.

Table 1: Priority Temperature Coefficients

| Point2DLargeEnv | Reacher | FetchReach | SawyerReach |
|:---:|:---:|:---:|:---:|
| 9.0 | 9.0 | 9.0 | 7.0 |

| HandManipulatePenRotate | HandReach | SawyerDoor |
|:---:|:---:|:---:|
| 7.0 | 7.0 | 1.0 |

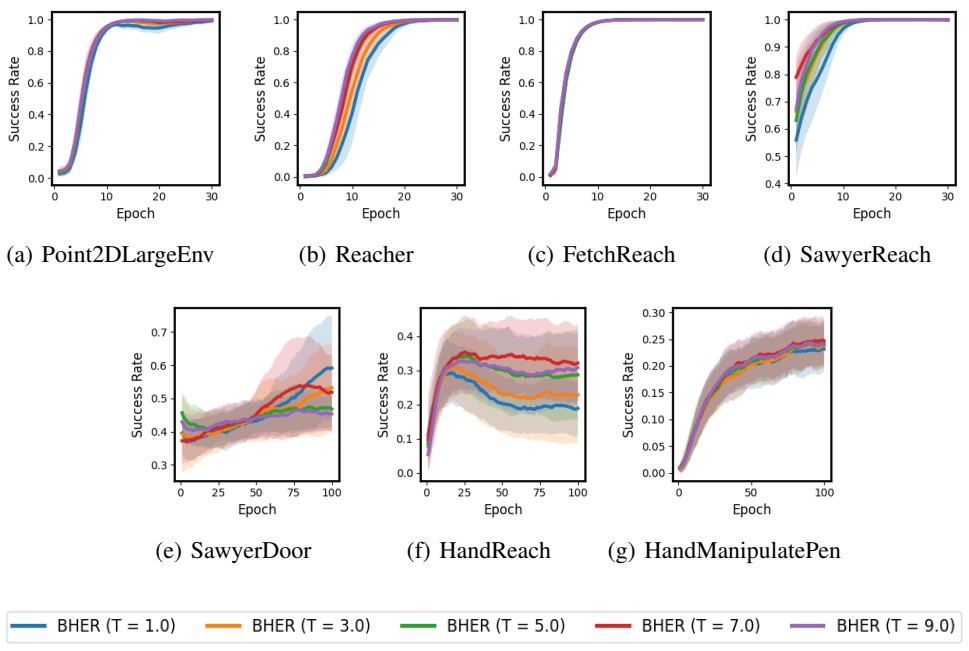

Figure 6: Performance of BHER with different priority temperature coefficient.

We experimented with five priority temperature coefficients of $\{1.0, 3.0, 5.0, 7.0, 9.0\}$ in all environments. Fig. 6 shows the results of our comparative experiment. It can be seen that different environments have different optimal priority temperature coefficients. We select an optimal priority temperature coefficient for each environment, as shown in the Table 1.

## C ABLATION EXPERIMENT FOR ORIGINAL DATA

Section 4.2 shows that hindsight transitions have smaller variance, and learning small variance transitions is more conducive to improving the exploitation of agent. In other words, it is better to let the agent learn the hindsight transitions first.

Therefore, we design an ablation experiment to evaluate the influence of the original data on our algorithm. We modify our algorithm to not use the original data which we call BHER w/o original data. Fig. 6 shows that in some relatively simple environments, such as Point2DLargeEnv and FetchReach, not using original data has little effect on the performance of our algorithm. However, in other more complex environments, original data will seriously affect the performance of our algorithm. The ratio of the original data is not the focus of our paper, we set this ratio just like HER algorithm.

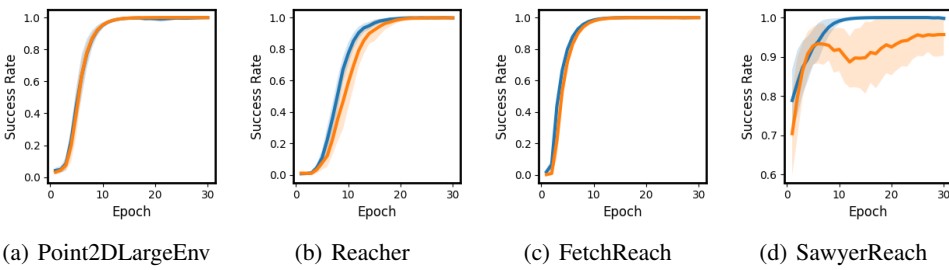

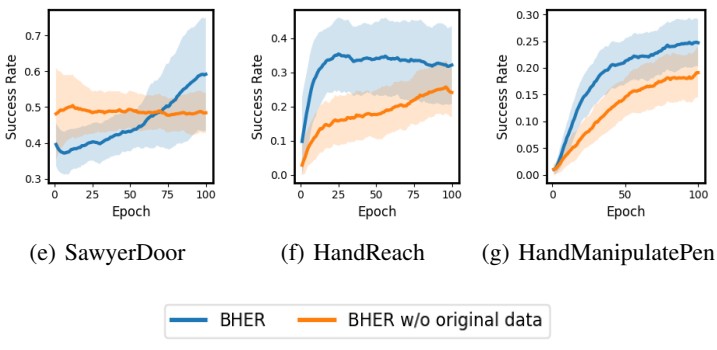

Figure 6: Ablation experiments results for original data.

We also evaluate the intuitive priority replay (higher priority for higher variance) without using the original data which we call BHER w/o original data w/ higher variance first. We don't evaluate this setting in all environments, because we think these results can already prove our method. Fig. 7 shows that even under the setting that does not use the original data, the counterintuitive priority replay method has better performance. This also illustrates the effectiveness of our counterintuitive prioritization, which improves the exploitation of agent.

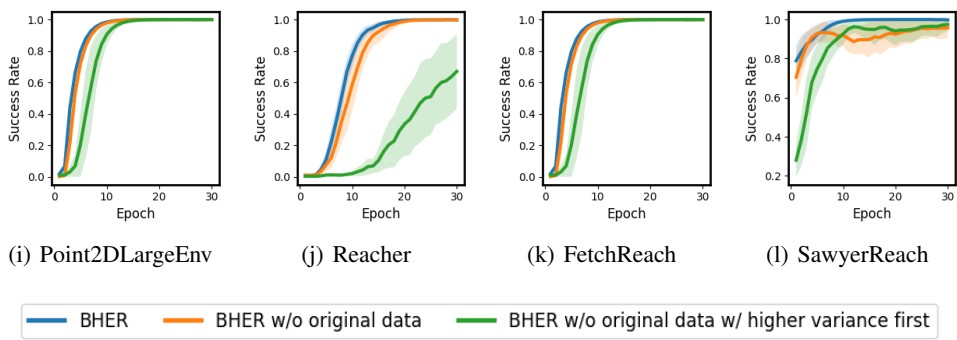

Figure 7: Ablation experiments results for original data and higher variance first.

## D    TRADE-OFF OF EXPLORATION AND EXPLOITATION IN TOY ENVIRONMENT

We use Reacher (for details, see Appendix A) as the toy environment to illustrate the trade-off between exploration and exploitation in our BHER algorithm. Fig. 8(a) and 8(b) shows that agent

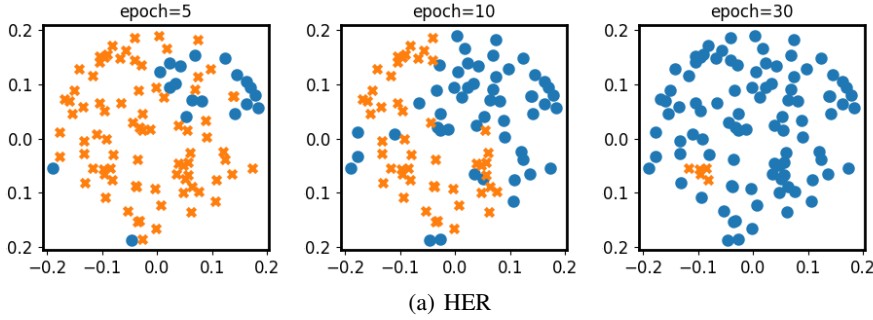

(a) HER

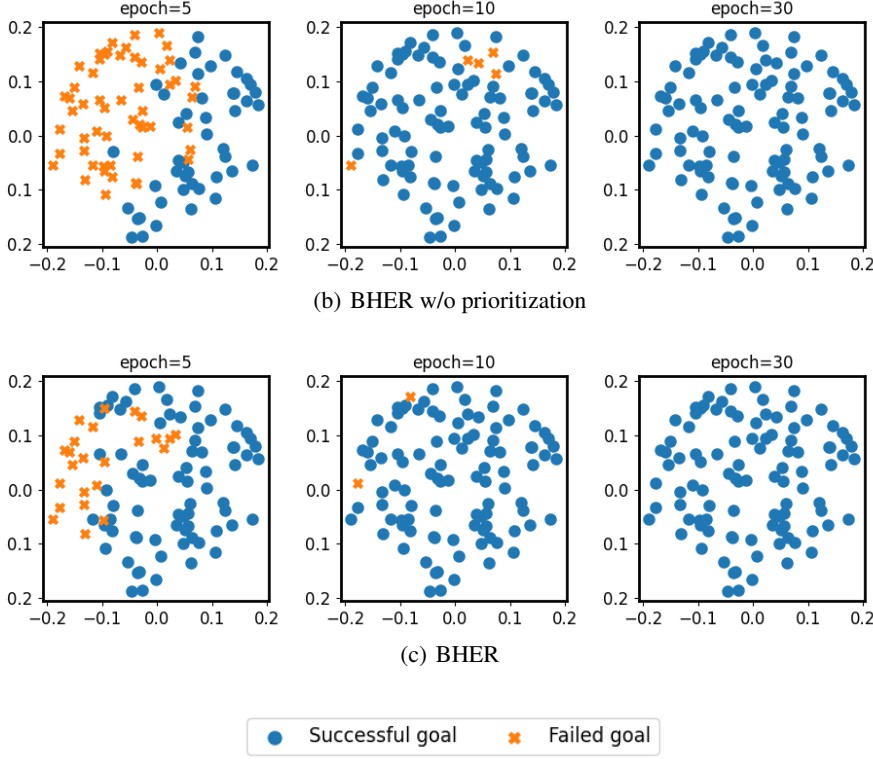

Figure 7: Distribution of the goals in Reacher. (BHER w/o prioritization only uses bootstrap principle on the basis of HER, but does not use any priority replay)

can explore more goals in a shorter period of time. This shows that bootstrap principle has improved the agent's exploration. Fig. 8(b) and 8(c) illustrates that we can make the agent preferentially select more valuable goals to learn by using counterintuitive prioritization. This effectively improves the exploitation of agent, which thus learns the optimal strategy faster.

