# OpenReview forum: "Bootstrapped Hindsight Experience replay with Counterintuitive Prioritization"
_ICLR.cc/2022/Conference — ICLR 2022 Submitted_

### Official Review · Reviewer_PtA1 · 2021-11-01

**Correctness:** 4
**Technical Novelty And Significance:** 3
**Empirical Novelty And Significance:** 3
**Recommendation:** 5
**Confidence:** 4

**Main Review:**

## High level comments
### Strengths
* Incorporating the idea of multiple heads from bootstrapped DQN for goal-conditioned RL (GCRL) and then also using those multiple heads to also prioritize replay is a sensible extension of the idea to GCRL.
* The counterintuitive prioritization needed when using the variance of the Q-values for prioritization is an interesting effect that is evaluated empirically with a possible explanation that it balances exploration and exploitation. This empirical example comparing the variance in Q-values in HER vs the original samples is helpful to see why the lower variance samples should be helpful.

### Weaknesses:
* The paper focuses on this counterintuitive prioritization in the experiments. This focus is understandable since that is the novel aspect introduced in the paper. However, it is unclear whether the improved performance by BHER is due to the deep exploration due to multiple heads or this counterintuitive prioritization. The ablation that removes the prioritization seems to still perform almost as well, suggesting that most of the improvement is due to the improved exploration.

The above reason is why I am unconvinced about the experimental validation. Perhaps an additional ablation of BHER without the deep exploration would highlight the efficacy of the counterintuitive prioritization.

Also see below for questions about other comparisons.

## Detailed Comments

* Since the bootstrapped heads improve the exploration of the agent, I wonder how this exploration stacks up to other exploration techniques in this domain. For example, would it be better than the directed exploration induced by [1]? I understand that this paper is very recent, but comparison to some other exploration enhancing mechanisms would be good.
* The approach of using multiple heads and the variance in Q-values for picking samples seems related to [2]. Could authors clarify the difference, and why this method was not compared to in baselines?
* Section 4.2, which goes over the counterintuitive prioritization and why it is needed, is enlightening but also lacking. The comparison of the variance when using HER and when considering just samples from the environment show that HER samples have lower variance. The paper points out that this discrepancy could be due to the higher proportion of successful trajectories due to hindsight. This analysis is reasonable. But then the authors suggest that prioritizing higher variance samples would be similar to sampling without hindsight. My question is: What would happen if you only sampled higher variance samples from the relabeled experience, and sampled uniformly from the  true experience? Wouldn't that ensure that original samples are not sampled out of proportion?
* Secondly, by prioritizing lower variance samples, does BHER just sample more from relabeled experience?
* In the experiments, the Fetch robot task used for evaluation is just the Reach environment, similar to the Reacher task. Evaluation on some actual manipulation tasks such as Push or Pick and Place would be a better evaluation in this domain. This is a small nitpick, the other experiments deal with more challenging tasks. It just seems like the FetchReach task is not bringing much to the table, except nominally adding a domain.
* As mentioned earlier, in the ablations, BHER without deep exploration would be a good one to have. And in those figures, the baseline DDQN should also be plotted, to show the improvement.

## Minor Comments
* More preciseness in the problem setup (Section 3.1) would be appreciated. As it stands, this setup would be hard for someone not familiar with reinforcement learning literature to parse.
* Typo: Section 4.1, line 2


[1] - Adversarial Intrinsic Motivation for Reinforcement Learning, Durugkar et al., 2021
[2] - Automatic Curriculum Learning through Value Disagreement , Zhang et al., 2020

**Summary Of The Paper:**

This paper looks at the problem of goal-conditioned reinforcement learning, and focuses specifically on the problem of prioritizing replay in conjunction with hindsight experience replay (HER).
It does so by using the idea of multiple heads in the actor-critic policy and critic networks, inspired by bootstrapped DQN. These different heads are initialized separately, and are also used to induce deep exploration in the style of bootstrapped DQN.

The main idea is to use the variance in the q-values across the heads for different transitions when prioritizing transitions to use for learning. The algorithm, termed counterintuitive prioritization, prioritizes transitions with lower variance. An example on the reacher environment is used to illustrate why this counterintuitive prioritization assists learning, and hypothesizes that it is based on a tradeoff between exploration and exploitation.

Experiments are performed in two simpler environments (Point 2D and Reacher) and 5 environments simulating different robots (Fetch, Sawyer, and Hand). They compare with two algorithms that prioritize transitions for replay based on some other criteria, and show that the counterintuitive prioritization along with deep exploration due to multiple heads leads to faster improvement in success rate of reaching goals in terms of interactions with the environment.

Ablations try to tease apart whether this improvement is due to the improved exploration due to multiple heads or the better exploitation due to the counterintuitive prioritization.

**Summary Of The Review:**

The idea of enhancing exploration in goal-conditioned RL using bootstrapped DQN is reasonable. Also using the multiple heads to prioritize samples for replay is a good extension, which leads to the investigation in the paper.
The idea of counterintuitive prioritization is investigated using an empirical investigation that gives a good insight into the reasons why this approach can be expected to work.

However, the paper does not sufficiently tease apart the effect of prioritization versus the enhanced exploration.
The comparison to other techniques that prioritize their samples does not take into account the enhanced exploration either.
And the paper does not compare to other techniques that enhance exploration.

---

> ### Author Response · Authors · 2021-11-23
> **Response to reviewer PtA1**
>
> We sincerely appreciate your suggestions and comments. We provide point-by-point response below.
>
> **Question:** The improvement of multiple heads and counterintuitive prioritization.
>
> **Answer:** Bootstrapped principle has greatly improved the original HER algorithm.
> Based on this, we propose to further investigate the effect of different ways of prioritization.
> We agree that counterintuitive prioritization we currently implemented does not bring significant improvement in some of the environments. We conjecture that the distribution of the variances of the data does not scale large.
> We will design other measurements to verify this.
>
> **Question:** The difference with VDS.
>
> **Answer:** We think a big difference between our algorithm and the VDS algorithm is that VDS algorithm regards the ensemble of value functions as a separate module to measure the uncertainty of the data.
> The ensemble of value functions in VDS does not guide the agent to optimize policy, which means they does not guide the actor in DDPG. However, in our algorithm, the ensemble of value functions not only measure the uncertainty of the data, but also guide the corresponding actors to optimize policy.
> Another difference is that VDS only samples goal at the beginning of each episode as the desired goal, while our algorithm samples the transitions during the training process.
> We will consider VDS algorithm as a baseline and make a detailed analysis in the revision.
>
> **Question:** What would happen if it only sampled higher variance samples from the relabeled experience, and sampled uniformly from the true experience? Wouldn't that ensure that original samples are not sampled out of proportion?
>
> **Answer:** Thanks for the suggestion. We think one way to implement the mentioned sampling method is to calculate the average variance of the true data samples and use this average variance as a reference to assign all the true data points a uniform probability, which is calculated from the population of the other relabeled data. We will try this method in the revision.
> In Appendix C, Figure 7 shows that it is better to sample lower -variance samples than to sample higher-variance samples without true experience.
> We think this result also demonstrates the effectiveness of the counterintuitive prioritization.
>
> **Question:** Experiment on Fetch robot task.
>
> **Answer:** We will conduct experiments in more robot-controlled environments in the revision.
>
> **Question:** The lack of ablation without deep exploration.
>
> **Answer:**
> Bootstrapped principle allows the algorithm to make deep exploration, and it is also the precondition for our algorithm to calculate the priority.
> Therefore, we can't design an ablation experiment without deep exploration but with counterintuitive prioritization.
> In addition, we considered continuous control tasks in our experiments, so we did not compare with DDQN or other variant DQN algorithms.
> Per your suggestion, we will include discrete tasks and  implement counterintuitive prioritization through other ways in order to compare algorithm without deep exploration in the revision.

---

> > ### Comment · Reviewer_PtA1 · 2021-11-24
> > **Re: Response**
> >
> > I thank the authors for their response.
> > Since there have been no revisions to evaluate whether the effectiveness of the idea is due to the deep exploration or due to the prioritization, I am not revising my score.
> >
> > Some detailed responses below:
> >
> > > Bootstrapped principle has greatly improved the original HER algorithm. Based on this, we propose to further investigate the effect of different ways of prioritization.
> >
> > The above line of evaluation suggests that you should then compare other methods of exploration in goal-conditioned RL as well.
> >
> > > Bootstrapped principle allows the algorithm to make deep exploration, and it is also the precondition for our algorithm to calculate the priority. Therefore, we can't design an ablation experiment without deep exploration but with counterintuitive prioritization.
> >
> > You misunderstand my question. I am not asking you to not use bootstrapping, which is essential to your approach for prioritization. I am asking you to not use a different head for each episode to encourage deep exploration. Does the bootstrapping suffer if you do not do the deep exploration? Perhaps. But it is still one way you can evaluate your prioritization in isolation.

---

### Official Review · Reviewer_nX4W · 2021-11-01

**Correctness:** 2
**Technical Novelty And Significance:** 1
**Empirical Novelty And Significance:** 2
**Recommendation:** 1
**Confidence:** 4

**Main Review:**

### Quality
* #### Strengths
    * The authors seem to have picked up on something interesting in paying attention to how transitions are sampled during the course of HER. Empirically, their approach seems to provide reasonable to modest gains across various continuous control tasks.
* #### Weaknesses
    * ##### Major
        * The authors have some fundamental issues and misconceptions about the BootstrappedDQN (BootDQN) algorithm. Section 2 mentions that BootDQN addresses exploration by running multiple behavior policies in the environment; if so, then it would be true that the method would have no reason to solve sparse reward tasks a priori as described by the authors. However, the reason for BootDQN's success in exploration is because of posterior sampling [8] and the fact that the ensemble is an approximate posterior over the optimal action-value function of the MDP, conditioned on all agent interactions observed thus far. It is precisely this principle that allows the algorithm to address sparse-reward tasks [2]. The authors' description of Thompson sampling in BootDQN as an ad-hoc heuristic seems incorrect given the rigorous theoretical guarantees that accompany randomized least-squares value-iteration algorithms [3]. The authors are also confused about the use of replay buffers in BootDQN, claiming that data from each head is held separate. There is exactly one replay buffer used in BootDQN and Bernoulli masks are sampled and stored with each transition for implementing the statistical bootstrap. Overall, the connection the authors draw with BootDQN in this work seem rather disingenuous; the proposed algorithm is simply applying ensembles of actor-critic pairs with no connection to the statistical bootstrap, unlike BootDQN. Renaming the algorithm and rephrasing the contribution seem appropriate.
        * A more critical issue concerning BootDQN and the proposed BHER algorithm is that the former is a purely value-based RL algorithm that maintains a posterior distribution over the optimal action-value function. In contrast, the latter is an off-policy actor-critic algorithm where, naturally, the critic is meant to be an estimate of the action-value function induced by the actor policy. While the empirical results of this paper confirm empirical benefits of this ensembling heuristic (the ablation in Figure 5 shows that this is responsible for most of the BHER performance), the authors have offered no real justification for this ensemble actor-critic algorithm. What is the point of representing epistemic uncertainty over the actor and critic in this manner?
        * I don't find the so-called counterintuitive prioritization to be counterintuitive at all. It seems natural that hindsight transitions will serve the agent well only when there is little uncertainty in the associated optimal behavior under the relabeled goal. Can the authors explain why sampling based on higher variance seems to still maintain reasonable performance in three of seven environments shown in Figure 5?
	* ##### Minor
		* In the first paragraph of Section 4.2, the authors describe an instance of Prioritized Experience Replay (PER) [6] where the variance of the critic ensemble is used to prioritize transitions sampled from the replay buffer. They seem to confuse this technique with methods for intrinsic motivation [7] based on curiosity and Random Network Distillation.

### Clarity
* #### Strengths
    * The authors provide ample details about their experimental setup for reproducibility of their results.
* #### Weaknesses
    * ##### Major
        * Overall, the paper is not well written. There are numerous grammatical errors throughout, genuinely too many for me to sensibly list them all out here. Oftentimes, these errors are missing articles (for example, "it uses successful trajectories generated by agent as expert demonstrates") or incorrect phrases ("on the contrast", "we inference all the Q-values"). Normally, I wouldn't bother nitpicking at a small handful of these, but there are too many throughout the entire body for what may end up being a published conference paper.
	* ##### Minor
		* The authors should remove the phrase "importance sampling" that is used twice in the paper to, in my reading, talk about the importance of sampled goals, rather than the Monte-Carlo technique of the same name.

### Originality
* #### Strengths
    * The authors demonstrate a good instinct in examining how other techniques used in deep reinforcement learning might further improve the efficacy of HER.
* #### Weaknesses
    * ##### Major
        * Fundamentally, this paper rests on the idea of using ensembles and prioritized experience replay together, neither of which is new to (deep) reinforcement learning [1,4,5,9]. Though I do not know of any prior work that has explored this specific combination, it would not surprise me if such prior work already exists.
        * More importantly, there are other options for leveraging such ensembles that have not been addressed in this work [1,4]. Similarly, the authors only consider variance-based prioritization schemes, rather than the traditional prioritization based on TD-error or any recent variants of PER. Demonstrating that the authors' specific choices in the proposed approach are better than these existing approaches to ensembling and PER would dramatically improve what so far seems to be a rather incremental algorithm.
	* ##### Minor
		* While it is appropriate for the related work section to focus on HER, it should also acknowledge the two fundamental innovations of this paper (ensembles and prioritization schemes) and provide an overview of related work for these areas as well.

### Significance
* #### Strengths
    * The only positive I can glean from this paper are the empirical results which seem to support the use of ensembling in actor-critic algorithms. Figure 5 shows a marginal drop in performance when the proposed counterintuitive prioritization scheme is not used. That said, it is not clear that this paper advances our understanding of ensembling in deep RL any more than prior work.
* #### Weaknesses
    * ##### Major
        * Given the lack of comparisons mentioned above, it's difficult to assess how impactful the proposed approach will be. With the breadth of existing work on the topic, I'm unconvinced that this will add any novel insights into how practitioners use ensemble methods in reinforcement learning. The prioritization scheme, while slightly interesting on the surface, doesn't seem to be a critical ingredient to the proposed algorithm's success based on the ablation studies shown.
        * I don't believe any of the experiments have shown results for regular DDPG without the use of HER. Having this baseline in place is important as it communicates the extent to which HER is even necessary for achieving a reasonable level of performance in each of the examined environments.
	* ##### Minor
		*

# References
1. Lee, Kimin, Michael Laskin, Aravind Srinivas, and Pieter Abbeel. "Sunrise: A simple unified framework for ensemble learning in deep reinforcement learning." In International Conference on Machine Learning, pp. 6131-6141. PMLR, 2021.
2. Osband, Ian, and Benjamin Van Roy. "Why is posterior sampling better than optimism for reinforcement learning?." In International conference on machine learning, pp. 2701-2710. PMLR, 2017.
3. Osband, Ian, Benjamin Van Roy, Daniel J. Russo, and Zheng Wen. "Deep Exploration via Randomized Value Functions." J. Mach. Learn. Res. 20, no. 124 (2019): 1-62.
4. Peer, Oren, Chen Tessler, Nadav Merlis, and Ron Meir. "Ensemble Bootstrapping for Q-Learning." arXiv preprint arXiv:2103.00445 (2021).
5. Saphal, Rohan, Balaraman Ravindran, Dheevatsa Mudigere, Sasikant Avancha, and Bharat Kaul. "SEERL: Sample Efficient Ensemble Reinforcement Learning." In Proceedings of the 20th International Conference on Autonomous Agents and MultiAgent Systems, pp. 1100-1108. 2021.
6. Schaul, Tom, John Quan, Ioannis Antonoglou, and David Silver. "Prioritized Experience Replay." In ICLR, 2016.
7. Singh, Satinder, Richard L. Lewis, Andrew G. Barto, and Jonathan Sorg. "Intrinsically motivated reinforcement learning: An evolutionary perspective." IEEE Transactions on Autonomous Mental Development 2, no. 2 (2010): 70-82.
8. Strens, Malcolm. "A Bayesian framework for reinforcement learning." In ICML, vol. 2000, pp. 943-950. 2000.
9. Wiering, Marco A., and Hado Van Hasselt. "Ensemble algorithms in reinforcement learning." IEEE Transactions on Systems, Man, and Cybernetics, Part B (Cybernetics) 38, no. 4 (2008): 930-936.

**Summary Of The Paper:**

# Summary & Contributions
* The authors call attention to sparse-reward, goal-based tasks where Hindsight Experience Replay, through its provision of relabeled goals for otherwise failed experiences facilitates efficient learning.
* To further improve upon HER, the paper focuses on two algorithmic innovations: (1) maintaining multiple actor-critic pairs inspired by BootstrappedDQN and (2) applying a variant of prioritized experience replay that draws samples from the replay buffer inversely proportional to the current Q-value variance under the critic ensemble.
* The authors support their approach with comparative experiments to other variants of HER as well as ablation studies of their own proposed method.

**Summary Of The Review:**

# Final Remarks
* The authors summarize their contributions in Section 6. While their fifth point is true, I would respond to the other points as (1) there is actually no use or examination of the statistical bootstrap in this work as the ensembles rely solely on random initializations, BootDQN also does not distinguish data sources, and there are other principled means of addressing exploration through ensembles such as UCB (2) while the proposed prioritization is new, there are many others that also do not depend on the environment (TD-error) which have not been assesssed (3) the combination is largely uninteresting based on ablation studies in this work which show near negligible impact of the proposed prioritization scheme and (4) I don't believe DDPG (with or without HER) holds state-of-the-art for these Mujoco domains; I would suspect that lies with either Soft Actor Critic or TD3.

Taken together, I don't believe this paper is ready for publication at this time.



======= Post Rebuttal =======

I thank the authors for their response but the justifications for the utility of representing epistemic uncertainty and lack of baselines are shallow;  it's clear that substantial revisions are needed before the submission is ready for publication.

---

> ### Author Response · Authors · 2021-11-23
> **Response to reviewer nX4W**
>
> We sincerely appreciate your suggestions and comments. We provide point-by-point response below.
>
> **Question:** What is the point of representing epistemic uncertainty over the actor and critic in this manner?
>
> **Answer:** The uncertainty is measured over multi-head Q-values. Given state-action pairs, this uncertainty represents how well the transition has been visited and memorized by the agent.
>
> **Question:** Why sampling based on higher variance seems to still maintain reasonable performance in three of seven environments shown in Figure 5.
>
> **Answer:** We think in these tasks the variances of the relabeled data are relatively flat so that both the directions of prioritization do not show significant differences. We will provide more empirical analysis on this point.
>
>
> **Question:** Innovative in the way of combining the two prior methods.
>
> **Answer:** We think incorporating bootstrapped principle into HER with investigations of the way of using prioritization is new in the topic of HER research.
>
> **Question:** Other works for leveraging such ensembles that have not been addressed and the lack of comparisons.
>
> **Answer:** In our experiments, we mainly focused on the improvement over the standard HER algorithm in continuous control tasks, so we compared other variants of HER.
> We did not show the performance of the standard DDPG, because previous work has shown that standard DDPG without HER cannot handle these sparse rewards environment.
> Per your suggestion, we will compare our algorithm with other related algorithms with bootstrapped principle in the revision.

---

### Official Review · Reviewer_yX9d · 2021-11-01

**Correctness:** 4
**Technical Novelty And Significance:** 2
**Empirical Novelty And Significance:** 2
**Recommendation:** 5
**Confidence:** 2

**Main Review:**

Strengths:
- the idea is simple and easy to implement;
- the BHER can achieve a significant improvement across different tasks;

weakness:
- This paper claims that BHER can achieve a trade-off between exploration and exploitation about the hindsight experience, but there are not strong enough evidence (empirical or theoretical) to support this claim. Though the experiments in Appendix.D try to show this trade-off, the results can't convince me, Figure.8 only shows that the multiple head principle can have a better exploration about the goal than HER, and the BHER(multiple head principle with prioritization) have the best exploration about the goal, it doesn't show BHER scarifices the exploration so as to improve the exploitation;
- Most of the performance improvement is due to multiple head principle, the counterintuitive prioritization can only slightly improve the performance in most environments. So the reason behind the counterintuitive prioritization should be further investigated.

some questions and typos:
1. the caption of the last figure in Appendix should be 'Figure 8' instead os 'Figure 7';
2. About the experiment setting, this paper first said that the reward was sparse and non-negative, however, this is contrast to the setting that the agent gets a reward of 0 only when it reaches desired goal, otherwise it receives a reward of -1.


**Summary Of The Paper:**

This paper aims to improve learning efficiency for Hindsight Experience Replay(HER). Instead of learning with a fixed proportion of fake and original data like HER does, this paper proposes to adopt the multiple head structure used in Bootstrapped DQN and then utilize the uncertainty measured by the variance of multiple estimated Q-values so as to weight the prioritization of each transition. Specifically, the proposed BHER enhances the importance of data samples with lower uncertainty, and thus achieves a trade-off between exploration and exploitation, resulting in a higher sample efficiency.

**Summary Of The Review:**

The paper is built on the observation that  achieving different goals may need different pseduo success trajectories which are unfortunately not provided by the naive HER algorithm. The paper then provides a straightforward solution to this by adding a Boostrapped DQN onto HER so as to allow it explore deeper and be able to evaluate the goodness of a pseduo trajectory for different goals.  Although this idea is shown work well compared to the naive HER, I think that further more principled solution may be needed.

---

> ### Author Response · Authors · 2021-11-23
> **Response to reviewer yX9d**
>
> We sincerely appreciate your suggestions and comments. We provide point-by-point response below.
>
> **Question:** Explanation of trade-off between exploration and exploitation.
>
> **Answer:** We think the empirical evaluations on the priority temperature coefficient can help explanation of the exploration-exploitation tradeoff.
> When the priority temperature coefficient is larger, the priority distribution of the data is sharper, which indicates that the data with small variance of Q-value is more likely to be sampled. On the contrast, assigning higher priority to the data with smaller variance means that we are improving the exploitation of algorithm. We will find some other measurements depicting the tradeoff in our revision.
>
> **Question:** The improvement of counterintuitive prioritization.
>
> **Answer:** We agree that using counterintuitive prioritization does not bring much improvement in some environments, while we find that in Reacher-v2, the performance of the algorithm enhances a lot by incorproating counterintuitive prioritization. Moreover, we find that the opposite way of assigning higher priority to data with large variance deteriorates the performance much. Therefore, our main claim is that for HER methods, counterintuitive prioritization is more useful than the commonly considered prioritization. We will try to provide more explanations and empirical evaluations on this in our revision.

---

### Official Review · Reviewer_BnLV · 2021-11-02

**Correctness:** 3
**Technical Novelty And Significance:** 2
**Empirical Novelty And Significance:** 3
**Recommendation:** 3
**Confidence:** 4

**Main Review:**

**Significance**: Because the underlying ideas (bootstrap DQN, hindsight relabeling, and prioritized experience relay) are well known, the main criterion for evaluation is the empirical results. The results are quite strong. However, the complexity of the method and its sensitivity to hyperparameters make me a bit skeptical whether the paper will make a lasting impact on the field.

**Correctness**: I had two minor comments about the correctness of the paper:

* Much of the discussion of the paper discussed how hindsight relabeling helps with exploration. I don't think this is correct: exploration is a problem of data collection, while hindsight relabeling is a tool for performing updates using previously-collected data. Hindsight relabeling doesn't (directly) say anything about how the data should be collected.
* I didn't see bootstrap DQN as a baseline. As prior work has found that using multiple Q functions significantly improves performance (e.g., [1, 4]), I think this is an important baseline to add.

**Originality**: The proposed method is a new combination of old ingredients.

**Clarity**: The paper is mostly clear.
* I found the discussion of prioritization hard to follow. For example, I didn't understand what "higher variance first" means in Fig 5.
* The related work section lacks structure. I'd recommend organizing the related work thematically, and making sure that the commonality is clear.

**Minor comments**:

* "non-negative reward" -> "non-zero reward" (The agent could be getting negative dense rewards)
* "often fail to perform well in sparse rewards environments" -- Cite.
* "A brilliant ..." -- Cite [2], which predates the HER paper by ~25 years.
* "uniformly sampling one of them, as HER did" -- The HER paper actually tried many different sampling strategies, including strategies that sampled multiple goals. Check out the appendix of the HER paper.
* When discussing prior work in the introduction and in Sec. 4.1, I'd recommend citing [1, 4] and other recent papers that use multiple Q functions.
* "Both these methods don't work well ..." -- Cite.
* "Goal-Conditioned Supervised Learning" -- Cite [3], too.
* "the rewards in goal-conditioned environments are sparse" -- Actually define what the reward function is in an equation.
* "subtly applies" -> "applies". The application isn't subtle.
* I found Lines 5 and 8 of Alg 1 unclear.
* I found Fig 2 hard to interpret. Are there alternative ways of presenting the same data that more succinctly convey the same message?
* "sparse and non-negative ... reward of -1" -- Contradiction: the reward isn't non-negative if it is -1 for some states and actions.
* "acorss" -> "across"


[1] Lee, Kimin, et al. "Sunrise: A simple unified framework for ensemble learning in deep reinforcement learning." International Conference on Machine Learning. PMLR, 2021.

[2] Kaelbling, Leslie Pack. "Learning to achieve goals." IJCAI. 1993.

[3] Ding, Yiming, et al. "Goal-conditioned imitation learning." arXiv preprint arXiv:1906.05838 (2019).

[4] Chen, Xinyue, et al. "Randomized ensembled double q-learning: Learning fast without a model." arXiv preprint arXiv:2101.05982 (2021).

**Summary Of The Paper:**

This paper proposes a method for goal-conditioned RL that combines three ideas from prior work: bootstrap DQN, hindsight relabeling, and prioritized experience relay. Experiments show that the proposed method outperforms variants of hindsight relabeling.

**Summary Of The Review:**

While the empirical results of the paper are quite strong, the paper doesn't compare to a number of more recent methods that also use ensembles of Q functions. Moreover, the method is rather complex, and it doesn't provide a guiding explanation for why this particular combination of ideas is good.

------------------------
**After rebuttal**: Thanks to the authors for responding to some of the points raised in the review, and for running additional experiments. My two main concerns with the paper are (1) whether most of the empirical benefits are coming from the ensemble and (2) the clarity of the writing. While I am not convinced that the revised version addressed these concerns, I would encourage the authors to continue revising the paper and submit to a future conference.

---

> ### Author Response · Authors · 2021-11-23
> **Response to reviewer BnLV**
>
> We sincerely appreciate your suggestions and comments. We provide point-by-point response below.
>
> **Question:** 'Much of the discussion of the paper discussed how hindsight relabeling helps with exploration. I don't think this is correct: exploration is a problem of data collection...'
>
> **Answer:** We agree that 'exploration is a problem of data collection', and just because of this we think hindsight relabeling is a technique benefiting novel data collection. Normal exploration methods enhance the chance of seeing novel states. From our perspective, hindsight relabeling directly modifies the states (by noting that the goal is part of the state) such that the relabeled data turns out to be novel and the data has never been visited by the agent.Therefore, we believe that the process of hindsight relabeling is to collect novel data, thereby improving the exploration.
>
> **Question:** The lack of baselines.
>
> **Answer:** In our experiments, we considered continuous control tasks, so we did not compare with Bootstrapped DQN.
> Instead, we used Bootstrapped DDPG for experimental comparison. Although the bootstrapped principle improves the exploration of DDPG, it failed to perform well in goal-conditioned environments with sparse rewards. Similarly, we mainly focused on the improvement over the standard HER algorithm, so we compared other variants of HER.Per your suggestion, we will include discrete tasks and compare our algorithm with other related algorithms with bootstrapped principle in the revision.
>
> **Question:** The meaning of 'higher variance first'.
>
> **Answer:** The term 'higher variance first' indicates that we assign higher priority to the data with higher variance of Q-value.
> We will polish the writing stuffs in the revision.

---

> > ### Comment · Reviewer_BnLV · 2021-11-23
> > **Quick question**
> >
> > > Instead, we used Bootstrapped DDPG for experimental comparison.
> >
> > Great; this is exactly the comparison I was looking for! Where are the results for this baseline (which color line in which figure)? I (perhaps mistakenly) assumed that the BHER in Fig 4 referred to the proposed method, not the bootstrapped DDPG baseline.

---

> > > ### Author Response · Authors · 2021-11-24
> > > **Performance of Bootstrapped DDPG**
> > >
> > > We have used Bootstrapped DDPG for experimental comparison, but we didn't show it in the paper. Because the performance of Bootstrapped DDPG is very poor on Reacher-v2 which we used as a toy example, we did not experiment in a more difficult environment.
> > > But we put the results of experimental comparison in this link: https://drive.google.com/drive/folders/12gcR4nb7umRQIaPT44n3mEcrZleF8IzT?usp=sharing. 'BDDPG w/o prioritization' is the performance of Bootstrapped DDPG.
> > > It can be seen that in most environments, the performance of Bootstrapped DDPG is not as good as naive HER, and Bootstrapped DDPG does not work in more difficult environments.

---

> > > > ### Comment · Reviewer_BnLV · 2021-11-24
> > > > **Clarification about Bootstrapped DDPG comparison**
> > > >
> > > > Thanks for sharing these new results.
> > > > Does the "Bootstrapped DDPG" shown in these results use HER? What is the difference between these results and the "BHER w/o prioritization" in Figure 5?

---

> > > > > ### Author Response · Authors · 2021-11-25
> > > > > **Explanation of Bootstrapped DDPG comparison**
> > > > >
> > > > > Sorry for the confusion.
> > > > > "BDDPG w/o prioritization'' shown in new results only used bootstrapped principle on the basis of DDPG, but did not use HER, and did not use any prioritized replay.
> > > > > `"BHER w/o prioritization" shown in new results used bootstrapped principle  and HER on the basis of DDPG, but did not use any prioritized replay.
> > > > > "BHER w/o prioritization" is the same as shown in Figure 5.
> > > > > We just added performance curve of ``BDDPG w/o prioritization'' in new results.

---

### Decision · Program_Chairs · 2022-01-20

**Decision:**

Reject

**Comment:**

I thank the authors for their submission and active participation in the discussion. The reviewers unanimously agree that this submission has significant issues, including comparison to baselines/ablations [BnLV,yX9d,PtA1], clarity [BnLV], justification of the method [nX4W]. Thus, I am recommending rejection of this paper.